# New Records and Descriptions of Three New Species of *Quadriacanthus* (Monopisthocotyla: Dactylogyridae) from Catfishes (Teleostei: Siluriformes, Clariidae) in the Upper Congo Basin

**DOI:** 10.3390/ani15030395

**Published:** 2025-01-30

**Authors:** Gyrhaiss K. Kasembele, Maarten P. M. Vanhove, Archimède Mushagalusa Mulega, Auguste Chocha Manda, Michiel W. P. Jorissen, Wilmien J. Luus-Powell, Willem J. Smit, Charles F. Bilong Bilong, Dieu-ne-dort Bahanak

**Affiliations:** 1Unité de Recherche en Biodiversité et Exploitation durable des Zones Humides (BEZHU), Faculté des Sciences Agronomiques, Université de Lubumbashi, Lubumbashi P.O. Box 1825, Democratic Republic of the Congo; kapepula.kasembele@uhasselt.be (G.K.K.); augustechocha@gmail.com (A.C.M.); 2Research Group Zoology: Biodiversity & Toxicology, Centre for Environmental Sciences, Hasselt University, BE-3590 Diepenbeek, Belgium; archimedemulega@gmail.com (A.M.M.);; 3Department of Biology, Royal Museum for Central Africa, Leuvensesteenweg 13, BE-3080 Tervuren, Belgium; 4Laboratory of Biodiversity and Evolutionary Genomics, Department of Biology, KU Leuven, Ch. Deberiotstraat 32, BE-3000 Leuven, Belgium; 5Capacities for Biodiversity and Sustainable Development, Operational Directorate Natural Environment, Royal Belgian Institute of Natural Sciences, Vautierstraat 29, BE-1000 Brussels, Belgium; 6Laboratory Biodiversity, Ecology and Genome, Research Center Plant and Microbial Biotechnology, Biodiversity and Environment, Mohammed V University in Rabat, Rabat 10100, Morocco; 7Département de Biologie, Centre de Recherche en Hydrobiologie, Uvira P.O. Box 73, Democratic Republic of the Congo; 8DSI-NRF SARChI Chair in Ecosystem Health, Department of Biodiversity, University of Limpopo, Sovenga 0727, South Africa; wilmien.powell@ul.ac.za (W.J.L.-P.); willem.smit@ul.ac.za (W.J.S.); 9Laboratory of Parasitology and Ecology, Faculty of Sciences, University of Yaoundé 1, Yaoundé P.O. Box 337, Cameroon; bilong_bilong@yahoo.com; 10Institute of Agricultural Research-Minko Multipurpose Research Station, Meyomessala P.O. Box 167, Cameroon; dieunedortbahanak@gmail.com

**Keywords:** monogeneans, parasite, *Clarias*, fish, Africa, Bangweulu–Mweru, Lualaba

## Abstract

Monogenean flatworms are mainly parasitic in lower aquatic vertebrates including fish, anurans and chelonians. *Quadriacanthus* is one of the four genera infecting African clariids. To date, 45 members of the genus are described, but this is still embryonic compared to the expected parasite and host species richness in Africa. In this study, we examined the gills of five species of *Clarias*: *Clarias ngamensis*, *C. stappersii*, *C. buthupogon*, *C. gariepinus* and *C. theodorae.* Eight parasite species morphologically characterised as belonging to *Quadriacanthus* were identified, among them five known species and three that are newly described here. In view of the importance of the clariids in this study system, this part of aquatic biodiversity is still to be further studied to contribute to inventorying parasite species in Africa.

## 1. Introduction

In the past decade, there has been a growing focus on the study of monopisthocotylan parasites in the Upper Congo Basin. Several studies have been conducted on monopisthocotylan parasites in Lake Tanganyika and the Bangweulu–Mweru and Upper Lualaba Ecoregions [1] (e.g., [2,3,4,5,6,7,8,9,10,11,12]). Many of these studies focused on cichlid fishes, while Prudhoe [13], Vanhove et al. [14], Mushagalusa Mulega et al. [8] and Kasembele et al. [6] reported monopisthocotylans of clariid fishes, the former three for *Clarias gariepinus* (Burchell, 1822) and the latter for *C. ngamensis* Castelnau, 1861. Considering the whole Congo Basin, a total of 13 clariid species are reported [15]; among them, seven species are present in the Upper Congo Basin (*C. gariepinus*; *C. ngamensis*; *C. buthupogon* Sauvage, 1879; *C. dumerilii* Steindachner, 1866; *C. liocephalus* Boulenger, 1898; *C. stappersii* Boulenger, 1915; *C. theodorae* Weber, 1897) [16,17]. The latter five fish species have never been studied for their monopisthocotylan fauna [18]. Monopisthocotyla Brabec, Salomak, Kolísko, Scholz & Kuchta, 2023 is one of two major clades within Monogenea, which is a diverse but paraphyletic [19] group of parasitic flatworms reported from freshwater, brackish and marine fishes, crustaceans, cephalopods, amphibians, and reptiles with one species from a mammal [18]. The high species richness of Monopisthocotyla and the relatively narrow host specificity of its members are used today as an important asset in understanding parasite adaptation, evolution and speciation via host switching [20,21]. There is also a great interest in the host–parasite interaction networks due to the importance of considering the community context to better understand the ecological and evolutionary implications of these interactions [22]. In addition, monopisthocotylans can be used for detecting pollution, indicating host biology, and as tools in phylogeny, biogeography and host systematics [23,24,25,26].

African clariids harbour monopisthocotylan gill parasites belonging to *Birgiellus* Bilong Bilong, Nack and Euzet, 2007, *Gyrodactylus* von Nordmann, 1832, *Macrogyrodactylus* Malmberg, 1957 and *Quadriacanthus* Paperna, 1961 [27,28,29,30,31]. The latter genus comprises 45 known species worldwide recorded from clariids (13 fish species), bagrids (three fish species), notopterids (one fish species), claroteids (one fish species), and cichlids (one fish species) [6,18,32,33]. Given the large number of identified and unidentified host species not yet examined for parasites, it can be anticipated that the recorded monopisthocotylan diversity is incomplete. Thus, it can be hypothesised that further parasitological surveys in the Upper Congo Basin may lead to the recording of many parasite species, including undescribed species, as some fish species remain to be surveyed for their monopisthocotylan parasites (see e.g., [34]). This study focuses on the monopisthocotylan parasite fauna belonging to *Quadriacanthus* of five clariid fishes: the Blunt-toothed African catfish *C. ngamensis*, the Blotched catfish *C. stappersii*, *C. buthupogon*, the North African catfish *C. gariepinus* and the Snake catfish *C. theodorae* (Teleostei: Siluriformes, Clariidae) occurring in the Upper Congo Basin. Objectives include (i) inventorying the diversity of *Quadriacanthus* species and, in case of the discovery of new species, providing their description and (ii) analysing infection parameters of these monopisthocotylan parasites.

## 2. Materials and Methods

### 2.1. Study Area

This study was carried out in the south of the former Katanga province in the Upper Congo Basin, especially (I) in the Upper Lualaba Ecoregion (*sensu* Thieme et al. [1]) in the Lufira River, which is a major tributary of the Lualaba River [35,36], and (II) in the Bangweulu–Mweru Ecoregion (*sensu* Thieme et al. [1]) in (i) the Lubumbashi River, which originates west of the city of Lubumbashi and flows into (ii) the Kafubu River, which is a tributary of the Luapula River (Figure 1).

### 2.2. Fish Sampling

The sampling was opportunistic, without targeting the number of fishes to be dissected, in advance. Fish were bought alive from fishermen along the shores of Lufira (September 2015, November 2021), Lubumbashi and Kafubu rivers from November 2020 to July 2021 and then transported alive in aerated tanks containing river water to the laboratory of Biodiversité et Exploitation durable des Zones Humides (BEZHU) of the Université de Lubumbashi. Since gills of live fish cannot be exhaustively inspected for monogeneans [37], the fish were killed by severing the spinal cord just posterior to the cranium and identified following the keys proposed by Teugels [38] immediately prior to examination [39].

### 2.3. Parasite Sampling

Fish were dissected and right gill arches removed by dorso-ventral section. These were placed in a Petri dish containing water for examination using a stereomicroscope Optika 4.0.0 (OPTIKA Srl, Ponteranica, Italy). Parasites were dislodged from the gill filaments using entomological needles and fixed between slide and cover slip into a drop of ammonium picrate-glycerin, according to Nack et al. [40]. Twenty-four hours later, coverslips were sealed using nail varnish.

### 2.4. Monopisthocotylan Community Composition and Infection Parameters

Parasite identification was based on the morphology and the size of sclerotised parts of the genital and the haptor [20,40]. The measurements were carried out according to Gussev [41] and modifications by N’Douba et al. [42] (Figure 2) with the aid of a Leica DM 2500 microscope (Leica Microsystems CMS GmbH, Wetzlar, Germany), LAS software (3.8), and drawings of the sclerotised parts of the genital structures and haptor were made with the aid of Corel Draw Graphics Suite X8 software (Corel Corporation, www.corel.com/). Measurements in micrometers (µm) are presented as follows: mean (minimum–maximum). To comply with the regulations set out in article 8.5 of the amended 2012 version of the International Code of Zoological Nomenclature (ICZN) [43], details of the new species have been submitted to ZooBank. The Life Science Identifier (LSID) of the article is urn:lsid:zoobank.org:pub:FAF64692-C810-438A-9CDE-C3C796E3AF88. For each new species, the LSID is reported in the taxonomic summary. Note that the authors of the new taxa are different from the authors of this article; see Article 50.1 and Recommendation 50 A of the International Code of Zoological Nomenclature [44]. Parasite diversity is summarized by the species richness; regarding infection parameters, prevalence (P) and mean intensity (MI) are provided following definitions given by Margolis et al. [45] and Bush et al. [46] and categorised following Valtonen et al. [47].

## 3. Results

A total of 85 fish specimens of the five clariid fish species (*C. ngamensis*, *n* = 14; *C. stappersii*, *n =* 32; *C. buthupogon*, *n =* 9; *C. gariepinus*, *n =* 11; *C. theodorae*, *n =* 19) were dissected in the Upper Congo Basin (Table 1).

The investigation of gill filaments resulted in the record of eight monopisthocotylan species (Table 2). The morphology of the monogeneans found corresponds to the diagnosis of *Quadriacanthus* following Paperna [27] and amendments by Kritsky and Kulo [48]. Among these eight monopisthocotylan species recorded, five are known (*Q. aegypticus* El-Naggar & Serag, 1986, *Q. allobychowskiella* Paperna, 1979, *Q. amakaliae* Kasembele, Bahanak & Vanhove, 2024 (Figure 3a,b), *Q. domatanai* Kasembele, Bahanak & Vanhove, 2024 (Figure 3c,d), *Q. halajiani* Kasembele, Bahanak & Vanhove, 2024 (Figure 3e,f)) [6,30,49] and three (*Q. kalomboi* n. sp., *Q. bassocki* n. sp. and *Q. shigoleyae* n. sp.) are newly described (see Table 2; Figure 4, Figure 5, Figure 6 and Figure 7). Descriptions of new species are given below, and the infection parameters of all the retrieved species are provided in Table 2. No monopisthocotylan parasite was found on the gills of *C. stappersii* from Lubumbashi River nor from the gills of *C. theodorae* from both Lubumbashi and Kafubu rivers.

*Quadriacanthus kalomboi* n. sp.

ZooBank registration: The LSID for *Quadriacanthus kalomboi* Kasembele, Bahanak & Vanhove n. sp. is urn:lsid:zoobank.org:act:87B042BC-5206-4BCA-A207-227EF1226227.
Host: *Clarias stappersii*Locality: Democratic Republic of the Congo, Kafubu River 11°43′50.00″ S; 27°27′42.60″ E, G.K. Kasembele leg.Collection date: 4 June 2021Site of infection: gillsMaterial: The holotype HU n°1005 is deposited in the collection of the Research Group Zoology: Biodiversity & Toxicology of Hasselt University (Diepenbeek, Belgium).Prevalence: P = 8.3%; mean intensity: MI = 1Etymology: This species is named in honour of Kalombo Kabalika Clément, MSc, from the Unité de Recherche en Biodiversité et Exploitation durable des Zones Humides, Faculté des Sciences Agronomiques, Université de Lubumbashi, DR Congo, for his valuable and kind assistance during our field campaigns.
Description

Dorsal anchor without shaft nor guard, with broad base, shaft sharply curved, ending with a short point. Dorsal bar with rectangular centre, a medium median process posteriorly directed and two lateral expansions. Dorsal cuneus triangular. Ventral anchor without shaft nor guard with regularly curved blade. Ventral bar V-shaped with two lateral branches. Y-shaped ventral cuneus. Seven pairs of hooks: pair IV larger than the rest which are almost equal in length (Figure 4 and Figure 7a,b). Tubular male copulatory organ (MCO) in the form of straight tube, flared, wide at the base and narrow at the distal extremity. Accessory piece (AP) ending like a hook with three small outgrowths: two posteriorly and one anteriorly, giving an appearance of a flame. No sclerotised vagina observed. Measurements of sclerotised pieces taken from one flattened specimen are shown in Table 3.

**Figure 4 animals-15-00395-f004:**
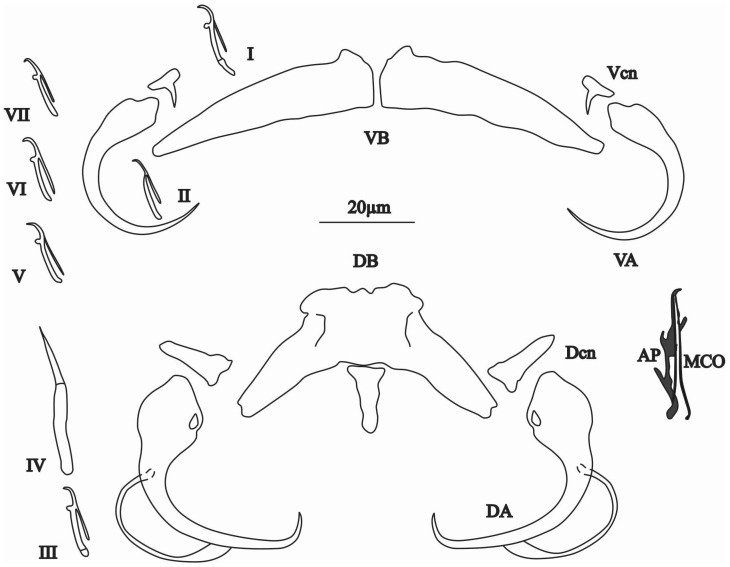
Sclerotised parts of the genitals and haptor of *Quadriacanthus kalomboi* n. sp. with the male copulatory organ (MCO), accessory piece (AP), ventral bar (VB), ventral anchor (VA), ventral cuneus (Vcn), dorsal bar (DB), dorsal anchor (DA), dorsal cuneus (Dcn), (I–VII) hooks.

Differential diagnosis

This species is comparable to *Q. lubandaensis* Kasembele, Bahanak & Vanhove, 2024 and *Q. fornicatus* Francová & Řehulková, 2017, described from *C. ngamensis* in the Bangweulu–Mweru Ecoregion (DRC), and *C. gariepinus* from Sudan [6,50], respectively, because of the morphology of the ventral bars (shape of the two elongated components) and anchors (moderate base, curved shaft, long point) and the MCO (straight tube, base simple). However, *Q. kalomboi* n. sp. can be differentiated from these two species by the morphology of its accessory piece in the shape of a flame with three small outgrowths versus in the form of a spike-like structure for *Q. fornicatus* and a simple hook-like ending in *Q. lubandaensis*.

*Quadriacanthus bassocki* n. sp.

ZooBank registration: The LSID for *Quadriacanthus bassocki* Kasembele, Bahanak & Vanhove n. sp. is urn:lsid:zoobank.org:act:5C1B53FD-FE53-4F70-A354-76EC47FDF0E7.
Host: *Clarias gariepinus*Locality: Democratic Republic of the Congo, Kafubu River 11°43′50.00″ S; 27°27′42.60″ E, G.K. Kasembele leg.Collection date: 4 June 2021Site of infection: gillsOther hosts: *C. stappersii*; *C. buthupogon*Other locality: DR Congo, Lubumbashi River 11°39′19.20″ S; 27°27′37.40″ EMaterial: The holotype HU n°1006 and 36 paratypes HU n°1007–1042 are deposited in the collection of the Research Group Zoology: Biodiversity & Toxicology of Hasselt University (Diepenbeek, Belgium).Prevalence: P = 50% (Kafubu River); mean intensity: MI = 1Etymology: This species is named in honour of Dr. Etienne Bassock Bayiha of the Laboratory of Parasitology and Ecology, Faculty of Sciences, University of Yaoundé 1, Cameroon, for his contribution in lab work.
Description

Dorsal bar with rectangular centre, a long median process posteriorly directed and two lateral expansions. Dorsal anchor without shaft nor guard with broad base, shaft sharply curved, ending with a short point. Dorsal cuneus triangular. Ventral bar V-shaped with two lateral branches. Ventral anchor without shaft nor guard, with base smaller than the base of the dorsal anchor, and curved blade. Ventral cuneus smaller than dorsal one. Seven pairs of hooks: pair IV longer than the rest, which are almost equal in length (Figure 5 and Figure 7c,d). A tube-shape MCO flared, wide and thick-walled at its base, gradually narrowing towards the distal extremity. The accessory piece, simple and articulated to the MCO, ending in a well-developed long point. Tubular vagina partly sclerotised. Measurements of sclerotised pieces taken from ten flattened specimens are shown in Table 3.

**Figure 5 animals-15-00395-f005:**
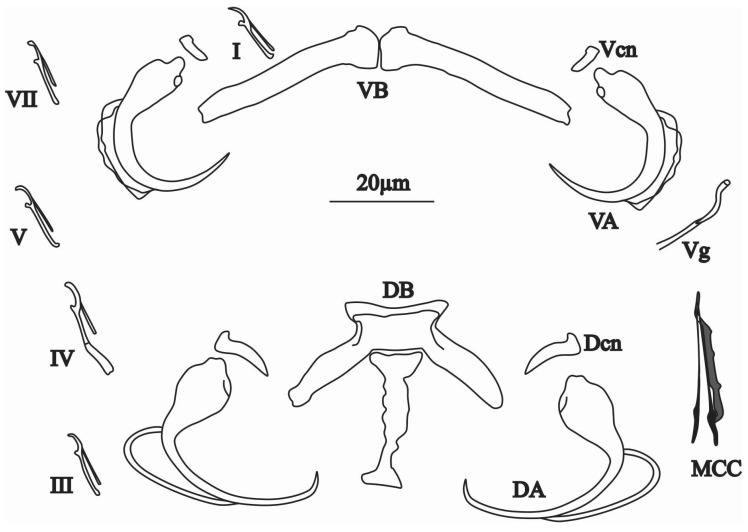
Sclerotised parts of the genitals and haptor of *Quadriacanthus bassocki* n. sp. with the male copulatory organ (MCO), accessory piece (AP), vagina (Vg), ventral bar (VB), ventral anchor (VA), ventral cuneus (Vcn), dorsal bar (DB), dorsal anchor (DA), dorsal cuneus (Dcn), (I–VII) hooks.

Differential diagnosis

*Quadriacanthus bassocki* n. sp. is comparable to *Q. amakaliae*, *Q. barombiensis* Bahanak, Nack & Pariselle, 2022, *Q. levequei* Birgi, 1988 and *Q. anaspidoglanii* Akoumba, Pariselle & Tombi, 2017, described, respectively, from *C. ngamensis* (in DR Congo)*, C. maclareni* Trewavas, 1962, *C. pachynema* Boulenger, 1903 and *Notoglanidium macrostoma* (Pellegrin, 1909)*,* in Cameroon [20,51,52]. They are similar in the morphology of the dorsal bar with a rectangular centre, a funnel-shaped median process that is posteriorly directed, without filaments at its end like in *Q. amakaliae* and *Q. anaspidoglanii*; the tubular shape of the MCO, wide at its base and gradually shrinking towards the distal extremity; the shape of the distal part of the AP ending in a point like in *Q. levequei*, *Q. anaspidoglanii* and *Q. barombiensis.* However, *Q. bassocki* n. sp. can be differentiated from *Q. barombiensis* and *Q. levequei* by the absence of filaments on the median process, which is present in the latter two parasite species; the distal extremity of the AP ending in a simple point for *Q. bassocki* n. sp. *versus* the distal extremity of the AP surrounded by two filaments for *Q. amakaliae* and two small hooks for *Q. levequei.*

*Quadriacanthus shigoleyae* n. sp.

ZooBank registration: The LSID for *Quadriacanthus shigoleyae* Kasembele, Bahanak & Vanhove n. sp. is urn:lsid:zoobank.org:act:99AD47E2-D042-4125-A297-029295CD1CE2.
Host: *Clarias ngamensis*Locality: Democratic Republic of the Congo, Lufira River 11°4′31.60″ S; 26°55′2.40″ E, G.K. Kasembele leg.Collection date: 30 September 2015Site of infection: gillsMaterial: The holotype HU n°1043 and 48 paratypes HU n°1044–1091 are deposited in the collection of the Research Group Zoology: Biodiversity & Toxicology of Hasselt University (Diepenbeek, Belgium).Prevalence: P = 33.3%; mean intensity: MI = 17 ± 20.7 parasitesEtymology: This species is named in honour of Miriam Isoyi Shigoley, MSc, for her kind contribution in the lab work, during the first author’s predoctoral visit at the Centre for Environmental Sciences of Hasselt University (Belgium).
Description

Dorsal anchor without shaft nor guard with broad base, shaft sharply bent, short point. Dorsal bar with rectangular centre, a broad and long median process posteriorly directed and two lateral expansions. Dorsal cuneus triangular. Ventral anchor without shaft nor guard with narrow base and curved blade. Ventral bar V-shaped with two lateral branches. Ventral cuneus triangular. Seven pairs of hooks: pair IV larger than pairs III, I, V, VI and VII, the four latter pairs almost equal in length (Figure 6 and Figure 7e,f). MCO tube-shaped, flared at the base and distal end like a femur (thighbone). Accessory piece, articulated around the MCO, composed of narrow base, thick median part, an animal tail-like distal part and ending in a point. No sclerotised vagina observed. Measurements of sclerotised pieces taken from seven flattened specimens are shown in Table 3.

**Figure 6 animals-15-00395-f006:**
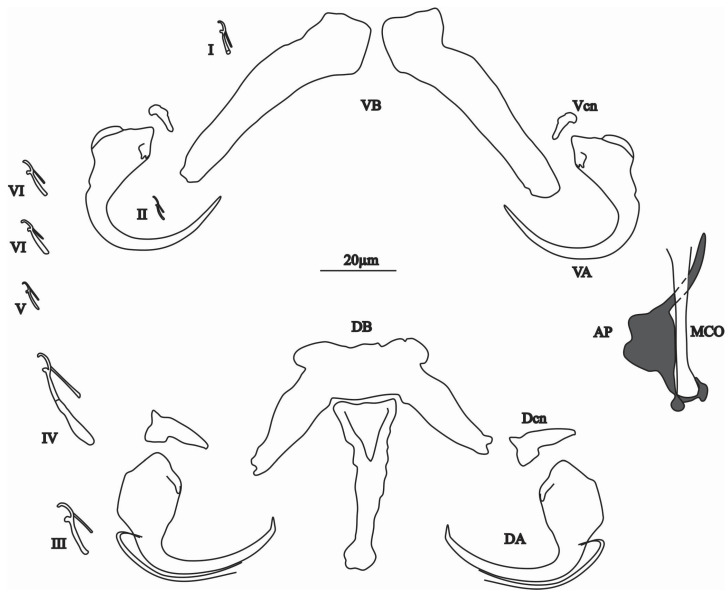
Sclerotised parts of the genital and haptor of *Quadriacanthus shigoleyae* n. sp. with the male copulatory organ (MCO), accessory piece (AP), ventral bar (VB), ventral anchor (VA), ventral cuneus (Vcn), dorsal bar (DB), dorsal anchor (DA), dorsal cuneus (Dcn), (I–VII) hooks.

Differential diagnosis

*Quadriacanthus shigoleyae* n. sp. resembles *Q. curvicirrus* Kasembele, Bahanak & Vanhove, 2024 described from *C. ngamensis* (in DR Congo) [6]. They are similar in the morphology of the ventral (shape of the two elongated branches) and dorsal (with rectangular centre, broad and long median process posteriorly directed and two lateral expansions) bars, and the dorsal anchors (broad base, shaft sharply bent, short point) and cunei (triangular). They can be distinguished by the characteristic genital shape of each species, femoral tube-shaped MCO, with a thickened AP ending in a tail-shaped extremity with a pointed ending for *Q. shigoleyae* n. sp. versus a tube-shaped MCO curved at its distal end with a simple base and a massive accessory piece with outgrowths posteriorly and anteriorly directed for *Q. curvicirrus.*

**Figure 7 animals-15-00395-f007:**
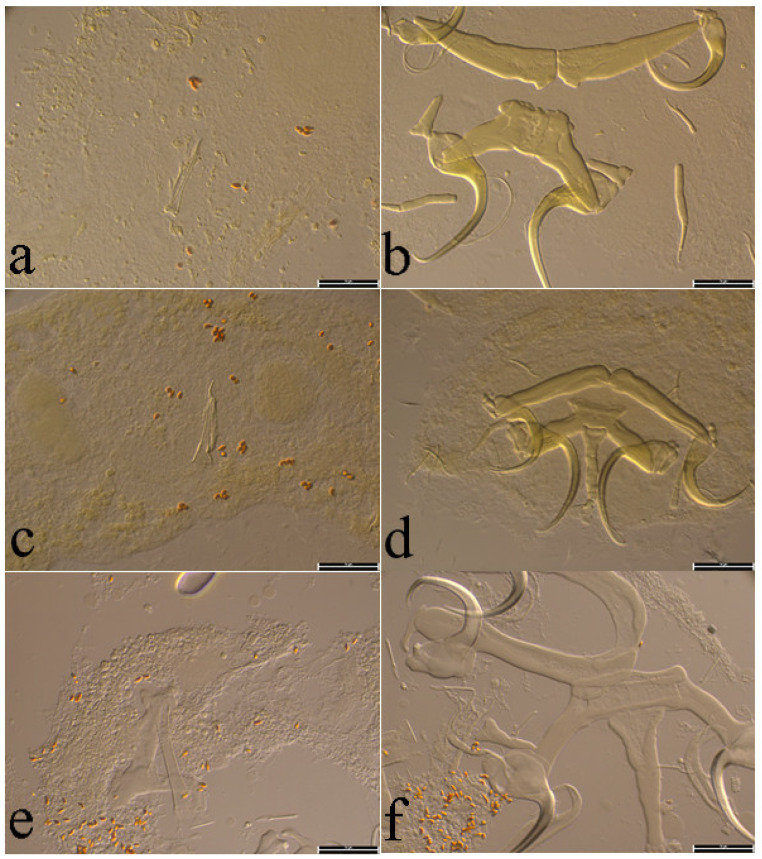
Photomicrographs of the sclerotised structures of the genitals and haptor of (**a**,**b**) *Quadriacanthus kalomboi* n. sp. ex *Clarias stappersii* from Kafubu River, (**c**,**d**) *Quadriacanthus bassocki* n. sp. ex *Clarias buthupogon* from Lubumbashi River, and (**e**,**f**) *Quadriacanthus shigoleyae* n. sp. ex *Clarias ngamensis* from Lufira River. Scale bar 20µm.

## 4. Discussion

The new records of *Q. aegypticus*, *Q. allobychowskiella*, *Q. amakaliae*, *Q. domatanai*, *Q. halajiani* and the descriptions of *Q. kalomboi* n. sp., *Q. bassocki* n. sp. and *Q. shigoleyae* n. sp. increase the knowledge of the diversity of monopisthocotylans in the Upper Congo Basin. Consequently, the number of known species of *Quadriacanthus* is extended to 48. From Africa, Řehulková et al. [18] reported 34 valid species; Bouah et al. [53] described two species; Bahanak et al. [20] described one species; Mushagalusa Mulega et al. [9] described one species; Kasembele et al. [6] described five species; and from Asia, Tripathi et al. [54] redescribed and synonymised two species of *Quadriacanthus* in their checklist. This study is the first record of monogenean flatworms on *C. stappersii* and *C. buthupogon.* Given the sample size for all fish species (including *C. gariepinus*, *C. ngamensis* and *C. theodorae*), it cannot be ascertained that the number of parasite species recorded is exhaustive.

The discovery of several new species and new host records in this study further emphasises how understudied this group is.

Regarding host range, *Q. amakaliae*, which is originally described from *C. ngamensis* in Lake Lubanda (DR Congo) [6] and now recorded on *C. stappersii* and *C. buthupogon*, should currently be considered a parasite with a mesostenoxenous specificity (more than one host, but restricted to one genus), infesting two or more congeneric host species [55,56,57]. The record of *Quadriacanthus aegypticus*, *Q. allobychowskiella, Q. domatanai* and *Q. halajiani* in this study is considered as a new locality record given that these parasite species were already reported on *C. ngamensis* by Kasembele et al. [6]. Another noteworthy observation is the description of *Q. bassocki* n. sp. from *C. gariepinus* which is known to harbour 11 species of *Quadriacanthus* from different basins/ecoregions or geographic areas [18,32,33,54], bringing the number of species within the genus infesting it to 12. This study demonstrates again that more sampling efforts can lead to the discovery of more parasite species and the recording of more host–parasite combinations even on relatively better-studied host species. Combes [58] states that sampling fewer than 30 hosts does not reveal the presence of species with low prevalence and abundance.

Indeed, differences in infection parameters could be attributed to factors such as season, biogeographical distribution, sample size or other environmental conditions. Communities of monogeneans have been shown to vary seasonally and across different habitat types, and the composition of parasite species may differ between regions and basins [59,60]. Nevertheless, the record of species being random, other similar studies successfully investigated less than 30 host specimens, and they still recorded or described many parasite species including rare ones, e.g., *Q. aegypticus* (P = 6.7%, MI: 1 ± 1) and *Q. amakaliae* (P = 6.7%, MI: 1 ± 0) recorded by [5] on *C. ngamensis* in Lake Lubanda (n, number of fish specimens= 15) and the Luapula River (*n =* 15); *Q. thysi* N’Douba, Lambert & Euzet 1999 (P = 8.3%, MI = 0.08) infecting *Heterobranchus longifilis* Valenciennes, 1840 (*n =* 12) in Cameroon [61]; *Gyrodactylus nyingiae* (P = 33.3%, MI: 1 ± 0) on *Pterocapoeta maroccana* Günther, 1902 (*n =* 3) in Morocco [62]; and *Quadriacanthus* spp. (13.3% < P < 40%, 1 < MI < 5.3) from *C. gariepinus* (*n =* 15) in South Africa [33]. The results of this study are furthermore consistent with the hypothesis that the potential host diversity in the tropics and the relatively narrow specificity of *Quadriacanthus* spp. could lead to the discovery of several, also new, parasite species [51,52,63].

Considering infection parameters in the current study area, *Q. amakaliae* is the most prevalent species with P = 60% for *C. stappersii* reported from the Lufira River, P = 50% for *C. buthupogon* from the Kafubu River, which is followed by *Q. bassocki* n. sp. (P = 50%) on both *C. buthupogon* and *C. gariepinus* from the Kafubu River. Parasite taxa are classified as common (P > 50%), intermediate (10–50%), or rare (<10% prevalence) by Valtonen et al. [47]. The prevalence of the other *Quadriacanthus* species is 10 < P < 50% (rendering them intermediate taxa in this study system) except for *Q. kalomboi* n. sp. and *Q. bassocki* n. sp. both from *C. stappersii* from Kafubu River (rare taxa here, P < 10%). *Quadriacanthus amakaliae* was ranked as common taxon in the Lufira River system and intermediate taxon in the Kafubu River system here and had previously been recorded as rare taxon from *C. ngamensis* in the Upper Congo Basin [6]. Infection parameters change depending on host and environmental factors [64,65]. In terms of mean intensities, parasite species can be classified as being of high intensity (MI > 100), medium (50 ≤ MI ≤ 100), low (10 ≤ MI < 50) or very low intensity (MI < 10) [47]. Results of this study show two groups, the first comprising *Q. shigoleyae* n. sp. (MI = 17 ± 20.7) on *C. ngamensis* from Lufira River, *Q. bassocki* on *C. buthupogon* from Kafubu River (11 ± 0) and Lubumbashi River (8 ± 11.9), and *Q. amakaliae* (9.8 ± 7.7) on *C. stappersii* from Lufira River, showing low mean intensity in these systems. This is different from the second group comprising the rest of the species, here exhibiting very low mean intensity levels. Values found for the second group are similar to those found by Kasembele et al. [6] especially for *Q. aegypticus* and *Q. allobychowskiella* on *C. ngamensis* from the Upper Congo Basin. Once more, infection levels vary with environmental conditions (physico-chemical parameters of water, e.g., the state of pollution of the water, potential eutrophication), or the host densities. Particularly influential can be the difference between conditions in nature versus in aquaculture, where fish are close and in permanent contact, and an oncomiracidium can quickly find a host after a short free swimming period [24,65,66,67,68,69].

## 5. Conclusions

We reported eight gill monopisthocotylan species belonging to *Quadriacanthus* from the clariid fishes *C. ngamensis*, *C. stappersii, C. buthupogon* and *C. gariepinus* in the Upper Congo Basin. For *Q. amakaliae*, we report a new host record, and for *Q. aegypticus*, *Q. allobychowskiella, Q. domatanai* and *Q. halajiani*, we report new locality records. *Quadriacanthus kalomboi* n. sp., *Quadriacanthus bassocki* n. sp. and *Quadriacanthus shigoleyae* n. sp. are described as new species. For future sampling, higher numbers of host specimens and more regions in the Upper Congo Basin are intended to be covered to record more information on monopisthocotylan diversity [5,6].

## Figures and Tables

**Figure 1 animals-15-00395-f001:**
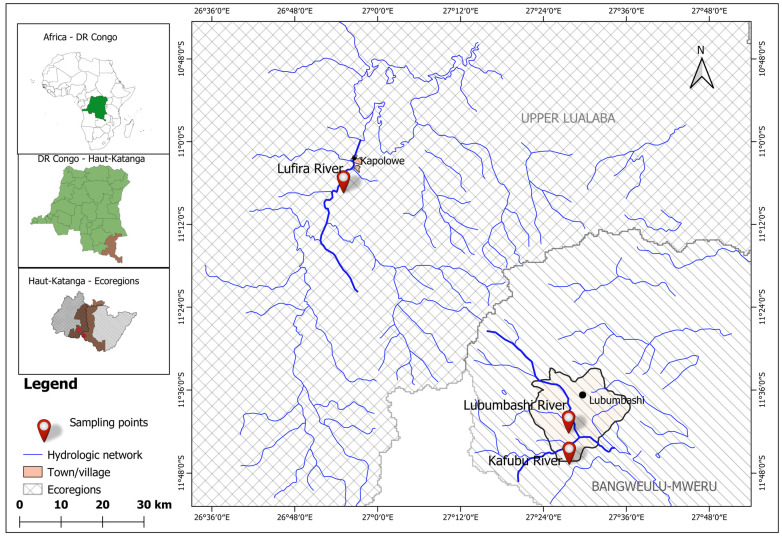
Map of sampling sites in the Upper Congo Basin: Lufira River 11°4′31.60″ S; 26°55′2.40″ E; Lubumbashi River 11°39′19.20″ S; 27°27′37.40″ E; Kafubu River 11°43′50.00″ S; 27°27′42.60″ E.

**Figure 2 animals-15-00395-f002:**
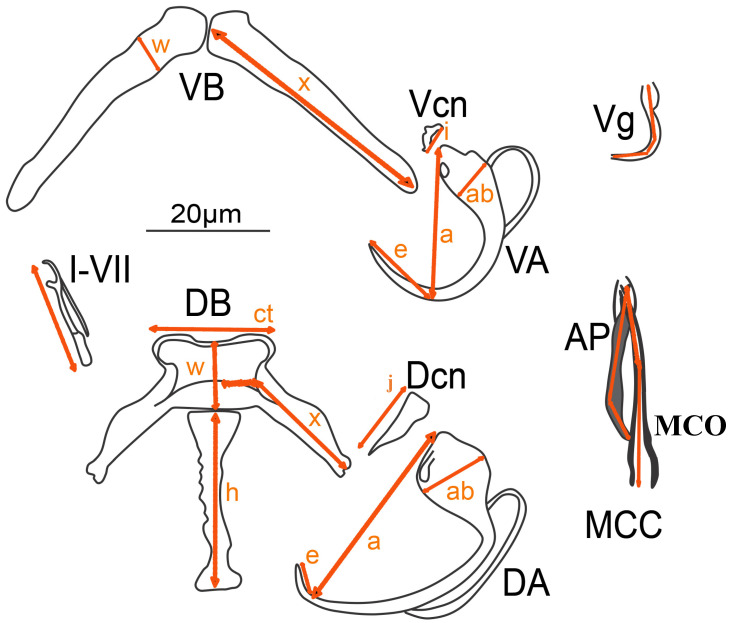
Morphometrics of *Quadriacanthus* spp. used in this study are based on Gussev [41] with modifications by N’Douba et al. [42]. MCO: male copulatory organ length; AP: accessory piece length; MCC: male copulatory complex (MCO and AP combined); I–VII: hook length; DB: dorsal bar: (x) length, (w) width, (h) median process length, (ct) centre length; DA: dorsal anchor: (a) length, (ab) base diameter or width, (e) point length; DCn: dorsal cuneus (j) length; VB: ventral bar: (x) length, (w) width; VA: ventral anchor: (a) length, (ab) base diameter or width, (e) point length; VCn: ventral cuneus (i) length; Vg: vagina length [6].

**Figure 3 animals-15-00395-f003:**
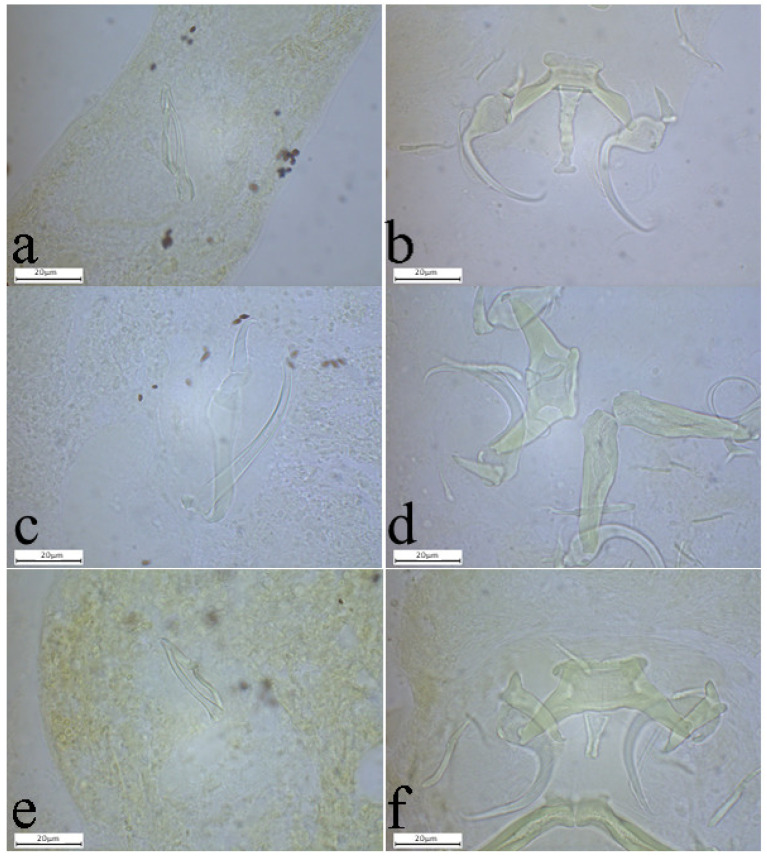
Photomicrographs of the sclerotised structures of the genitals and haptor of (**a**,**b**) *Quadriacanthus amakaliae* ex *Clarias stappersii* from Kafubu River, (**c**,**d**) *Quadriacanthus domatanai* ex *Clarias ngamensis* From Lubumbashi River, and (**e**,**f**) *Quadriacanthus halajiani* ex *Clarias ngamensis* from Kafubu River. Scale bar: 20 µm.

**Table 1 animals-15-00395-t001:** The number of specimens per fish species per sampling site.

Fish Species	Sampling Site		
	Lufira River	Lubumbashi River	Kafubu River
*Clarias ngamensis*	6	3	5
*C. stappersii*	10	10	12
*C. buthupogon*	-	7	2
*C. gariepinus*	-	7	4
*C. theodorae*	-	11	8

**Table 2 animals-15-00395-t002:** The monopisthocotylan parasite species recovered from *Clarias ngamensis*, *Clarias stappersii*, *Clarias buthupogon* and *Clarias gariepinus* from the Upper Congo Basin. The different localities are represented by the following abbreviations: Luf: Lufira River; Lub: Lubumbashi River; Kaf: Kafubu River.

Host Species	Parasite Species	Locality	No. of Examined Hosts	No. of Infested Hosts	Mean Intensity
*C. ngamensis*	*Quadriacanthus shigoleyae* n. sp.	Luf	6	2	17
	*Q. aegypticus*	Lub	3	1	1
		Kaf	5	1	1
	*Q. allobychowskiella*	Lub	3	1	1
		Kaf	5	2	1.5
	*Q. domatanai*	Lub	3	1	2
		Kaf	5	1	1
	*Q. halajiani*	Kaf	5	1	1
*C. stappersii*	*Q. amakaliae*	Luf	10	6	9.8
		Kaf	12	6	2.8
	*Q. kalomboi* n. sp.	Kaf	12	1	1
	*Q. bassocki* n. sp.	Kaf	12	1	2
*C. buthupogon*	*Q. amakaliae*	Lub	7	2	3
		Kaf	2	1	1
	*Q. bassocki* n. sp.	Lub	7	2	8
		Kaf	2	1	11
*C. gariepinus*	*Q. bassocki* n. sp.	Lub	7	1	6
	*Q. bassocki* n. sp.	Kaf	4	2	1

**Table 3 animals-15-00395-t003:** Obtained measurements (in micrometers) of *Quadriacanthus* species. Legend: MCO: male copulatory organ length; AP: accessory piece length; I–VII: hook length; DB: dorsal bar: (x) length, (w) width, (h) median process length, (ct) centre length; DA: dorsal anchor: (a) length, (ab) base diameter or width, (e) point length; DCn: dorsal cuneus length; VB: ventral bar: (x) length, (w) width; VA: ventral anchor: (a) length, (ab) base diameter, (e) point length; VCn: ventral cuneus length. Number of flattened specimens on which measurements of sclerotised pieces were taken: (n).

	*Quadriacanthus kalomboi* n. sp. (*n =* 1)	*Q. bassocki* n. sp. (*n =* 10)	*Q. shigoleyae* n. sp. (*n =* 7)
MCO	24.16	24.8 (16–29.8)	45.3 (38.6–50.7)
AP	24.14	24.1 (18.9–27.6)	48.3 (43.5–51.3)
I	15.44	12 (10.7–12.8)	15.4 (14.5–16.3)
II	11.4	11.3 (7.7–12.1)	12.1 (10.8–13.4)
III	14.42	12.7 (11.5–13.5)	13.9 (12.4–15.1)
IV	30.56	20.4 (19.6–21.3)	24.2 (21.8–26.9)
V	13.4	12.8 (12–13.5)	13.7 (13.3–14.1)
VI	13.44	12.9 (12.3–13.8)	13 (12.2–13.3)
VII	12.9	12.7 (11.8–13)	12.9 (12.4–13.1)
DBx	32.26	25.7 (22.7–27.6)	41.8 (39.9–43.4)
DBw	14.98	8.3 (7.5–9.7)	13.6 (11.3–15.2)
DBh	13.06	25.1 (23.2–27.2)	44.3 (41.5–47.6)
DBct	24.12	18.4 (16.8–21.2)	34.8 (30.7–38.5)
DAa	40.03	32.2 (23.5–35.4)	36.3 (34.8–38)
DAab	13.08	10.3 (9.9–10.8)	16.6 (16–17.1)
DAe	4.84	4.9 (3.8–13.6)	4.6 (3.7–5.3)
DCn	13.63	11.7 (10.8–13.6)	17.2 (15.9–18.3)
VBx	46.65	38.7 (35.5–41.3)	58.1 (40.4–63.1)
VBw	10.33	4.4 (4.1–6.3)	11.4 (10.3–12.4)
VAa	26.75	23.9 (23.3–24.8)	31.8 (30.9–32.5)
VAab	6.78	6.8 (6.2–7.3)	12 (10.2–13)
VAe	12.77	16.2 (15.5–17.2)	27.9 (27.2–28.8)
VCn	6.76	5.6 (5.3–6)	12 (10.8–14)

## Data Availability

Parasites were deposited in the collection of the Research Group Zoology: Biodiversity & Toxicology, at Hasselt University (Diepenbeek, Belgium) under accession numbers HU n°1005–HU n°1094 (type material) and XXII.3.45–XXIII.1.40 (reference material).

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
