# Peer review of "New Records and Descriptions of Three New Species of Quadriacanthus (Monopisthocotyla: Dactylogyridae) from Catfishes (Teleostei: Siluriformes, Clariidae) in the Upper Congo Basin"

_animals, 2025, doi:10.3390/ani15030395_

Round 1
Reviewer 1 Report (Previous Reviewer 3)
Comments and Suggestions for Authors
This is the second review of this manuscript. As it was noted in the first review, the article contains useful information on the occurrence of 8 species of monogeneans, including 3 new species, in a region poorly studied in this respect and is of scientific interest. The authors have taken into account my comments in the first review.
However, Table 1 has appeared in the present version of the paper. I see no need for it, as it simply repeats the ‘No. of hosts examined’ column in Table 2.
Also, what can the mean and error be when you only have 2 numbers in your sample, since 33% of 6 is 2 fish and 40% of 5 is also 2 fish, as is 28.6% of 7? Just write these 2 numbers in each case.
With these comments, the manuscript can be published in its present form.
Author Response
Comments and Suggestions for Authors
This is the second review of this manuscript. As it was noted in the first review, the article contains useful information on the occurrence of 8 species of monogeneans, including 3 new species, in a region poorly studied in this respect and is of scientific interest. The authors have taken into account my comments in the first review.
Response: We thank the reviewer for the valuable inputs on our manuscript. Responses to their concerns are given below.
Comment: However, Table 1 has appeared in the present version of the paper. I see no need for it, as it simply repeats the ‘No. of hosts examined’ column in Table 2.
Response: we have added this new table because we felt the need to be more explicit about the number of hosts included in this study, in view of questions raised by a reviewer and the editors in the previous round of reviewing. Therefore, we feel inclined to leave this table in the manuscript.
Table 1 also included, in contrast to Table 2, information on the hosts that did not yield parasites; e.g. Clarias stappersii examined from Lubumbashi River (Table 1) which was not infested, does not appear in table 2; C. theodorae from both Lubumbashi and Kafubu Rivers (Table 1), does not appear in table 2.
Comment: Also, what can the mean and error be when you only have 2 numbers in your sample, since 33% of 6 is 2 fish and 40% of 5 is also 2 fish, as is 28.6% of 7? Just write these 2 numbers in each
Response: To implement your remark, we now refrain from mentioning the standard deviation, and have replaced the term “prevalence” by “No of infested hosts”.
Reviewer 2 Report (Previous Reviewer 2)
Comments and Suggestions for Authors
Corretions and amendements are ok, no further comments.
Author Response
Comments and Suggestions for Authors: Corrections and amendments are ok, no further comments.
Response: We thank the reviewer for the revision of our manuscript.
Reviewer 3 Report (Previous Reviewer 1)
Comments and Suggestions for Authors
All the suggestions made were accepted. The article, although simple and without biotechnological tools, should be published.
Author Response
Comments and Suggestions for Authors: All the suggestions made were accepted. The article, although simple and without biotechnological tools, should be published.
Response: We thank the reviewer for the revision of our manuscript.
This manuscript is a resubmission of an earlier submission. The following is a list of the peer review reports and author responses from that submission.
Round 1
Reviewer 1 Report
Comments and Suggestions for Authors
New records and descriptions of three new species of Quadriacanthus (Monopisthocotyla: Dactylogyridae) from catfishes (Teleostei: Siluriformes, Clariidae) in the Upper Congo basin
In the abstract, you should put a short introductory sentence before the objective.
All keywords should be different from the title.
The introduction should be expanded to show more about the importance of these parasite species and their limitations for fish.
In the material and methods I didn't see the n of the samples. The work is classic and simple, but valid as long as it is well described.
Nowhere did I see any disclosure of an ethics committee.
Were any biotechnological tools used in the study?
Author Response
(i) In the abstract, you should put a short introductory sentence before the objective.
Response: Thank you, this has been added.
(ii) All keywords should be different from the title.
Response: Thank you, the keywords have been replaced.
- The introduction should be expanded to show more about the importance of these parasite species and their limitations for fish.
Response: Thank you, a paragraph has been added to show the importance of monopisthocotyleans in other research fields.
- In the material and methods I didn't see the n of the samples. The work is classic and simple, but valid as long as it is well described.
Response: The sampling was opportunistic, so we did not have a target number of fishes to be dissected, in advance. Hence, the number of fishes sampled (dissected) has been mentioned in the Table 1 in the Results section (for each fish species, per locality).
- Nowhere did I see any disclosure of an ethics committee.
Response: We provided to the Editor in Chief documents which are both authorizations for sampling, and the ethical approval for sampling. It was stated in the MS under section Institutional Review Board Statement: In the absence of relevant animal welfare regulations in the DRC, we used the guidelines and authorization in accordance with the Unité de Recherche en Biodiversité et Exploitation durable des Zones Humides (BEZHU) of the Université de Lubumbashi. Fishes were bought from fishermen, and references have been added (Autorisation d’échantillonnage N/Réf/fac/Agro/646/2015; Autorisation d’échantillonnage N/Réf/fac/Agro/714/2019).
- Were any biotechnological tools used in the study?
Response: No biotechnological tools used in the current study.
Reviewer 2 Report
Comments and Suggestions for Authors
Please check the numbering of the list of references. There are some inconstistencies and numbers in the text sometimes do not correspond with the numbers in the list of references.
Future research hopefully can increase numbers of fish to be examined for a better assessment and understanding of the ecological parameters.

Author Response
Comments: Please check the numbering of the list of references. There are some inconstistencies and numbers in the text sometimes do not correspond with the numbers in the list of references.
Response: Thank you, references have been checked and put in a correct way.
- Future research hopefully can increase numbers of fish to be examined for a better assessment and understanding of the ecological parameters.
Response: For sure, the more fish we dissect, the more informative are the results (especially for the ecological parameters). We have stated it in the conclusion: For future sampling, higher numbers of host specimens and more regions in the Upper Congo Basin are intended to be covered to record more information on monopisthocotylean diversity.
- Please check the numbering of the list of references. There are some inconsistencies.
Response: Thank you, references have been checked and put in a correct way.
Reviewer 3 Report
Comments and Suggestions for Authors
There is no doubt that the freshwater fauna of Africa, including parasites, is very diverse and understudied. This article provides new data on the distribution of 8 species of monogeneans, including 3 previously unknown to science. The study is certainly of scientific interest.
In general, the article can be accepted for publication.
I have only some minor remarks, which are as follows
1. Lines 51-52. The keywords need to be edited. Simply listing the species of parasite and host narrows the field of possible interest.
For example: parasites, monogeneans, Africa, river, fish.
2. Lines 57-58. Why complicate things, just give a link which means exactly that this information is given according to the article that is cited – “….and Upper Lualaba Ecoregions [1], e.g. [2-12].”
3. Lines 66-69. The authors of the taxon should be cited the first time it is mentioned. Monogenea without quotation marks. "The latter" is often repeated. Similarly, [19] should be used instead of (see Brabec et al. [19]).
«Monopisthocotyla Brabec, Salomak, Kolísko, Scholz & Kuchta, 2023 is one of the two major clades within the Monogenea, a diverse but paraphyletic [19] group of parasitic flatworms reported ….
- And check the numbering of the references!
4. Line 76. “… monopisthocotylean diversity is incomplete.” Monopisthocotylean is a member of the taxon Monopisthocotylea. Authors should be consistent; if you use an alternative taxonomy - Monopisthocotyla, use it everywhere.
5. Lines 127-129. It is NOT clear what the authors mean. What do they mean by ‘potentially unaccounted for’? Explain more clearly. Ten fish were examined, parasites were found in 5 fish, what could be ‘potentially unaccountable’?
6. Lines 140-141. ‘Diagnosis’ is applied to species, genus, family, etc., It is a generalisation of the main characteristics of all representatives of the corresponding taxon. Specimens do not have a diagnosis. It could be, ‘The morphology of the monogeneans found corresponds to the diagnosis of the genus...
7. Line 152. Table 1. Why do you write ‘minimal prevalence’? It's just a prevalence! In other samples, e.g., in other seasons or areas, it may be higher or lower. BUT here you are giving data for the particular sample studied. Why do you think that this level is "minimal"? Following your logic, you should then write ‘minimal mean intensity’. Correct it to ‘prevalence’.
8. Line 162. The scientific name of the host species should be given in Latin in the taxonomic summary here and below.
9. Line 169. Remove "minimal" here and below, leaving only "prevalence".
10. Lines 190-192. Confusion with reference numbers. If I read correctly, it is a reference to an unpublished article and a comparison with an unpublished species. The only thing left is to believe the authors.
11. Species descriptions are very brief. Usually, the structures of the haptor are described first and then the sexual system.
12. The "Taxonomic discussion" does not correspond to "discussion", it is very brief and more in the spirit of "Differential diagnosis".
Only qualitative, descriptive characters are used for species differentiation.
It is desirable to extend the differentiation by comparing the metric characters that differ significantly.
13. Throughout the text, make sure that the Latin names of the species are in italics.
14. Lines 354-360 repeat lines 362-369. Remove the repetition.
Author Response
There is no doubt that the freshwater fauna of Africa, including parasites, is very diverse and understudied. This article provides new data on the distribution of 8 species of monogeneans, including 3 previously unknown to science. The study is certainly of scientific interest. In general, the article can be accepted for publication. I have only some minor remarks, which are as follows:
- Lines 51-52. The keywords need to be edited. Simply listing the species of parasite and host narrows the field of possible interest. For example: parasites, monogeneans, Africa, river, fish.
Response: Thank you, keywords have been changed, and we think they are more general now.
- Lines 57-58. Why complicate things, just give a link which means exactly that this information is given according to the article that is cited – “….and Upper Lualaba Ecoregions [1], e.g. [2-12].”
Response: Thank you, we simplified this statement as suggested.
- Lines 66-69. The authors of the taxon should be cited the first time it is mentioned. Monogenea without quotation marks. "The latter" is often repeated. Similarly, [19] should be used instead of (see Brabec et al. [19]).
«Monopisthocotyla Brabec, Salomak, Kolísko, Scholz & Kuchta, 2023 is one of the two major clades within the Monogenea, a diverse but paraphyletic [19] group of parasitic flatworms reported ….
- And check the numbering of the references!
Response: Thank you, this has been rephrased accordingly. And numbering of the references has been checked.
- Line 76. “… monopisthocotylean diversity is incomplete.” Monopisthocotylean is a member of the taxon Monopisthocotylea. Authors should be consistent; if you use an alternative taxonomy - Monopisthocotyla, use it everywhere.
Response: Thank you, this has been changed.
- Lines 127-129.It is NOT clear what the authors mean. What do they mean by ‘potentially unaccounted for’? Explain more clearly. Ten fish were examined, parasites were found in 5 fish, what could be ‘potentially unaccountable’?
Response: This has been removed as we keep the term Prevalence (not Minimal prevalence).
- Lines 140-141.‘Diagnosis’ is applied to species, genus, family, etc., It is a generalisation of the main characteristics of all representatives of the corresponding taxon. Specimens do not have a diagnosis. It could be, ‘The morphology of the monogeneans found corresponds to the diagnosis of the genus...
Response: Thank you, this has been changed.
- Line 152.Table 1. Why do you write ‘minimal prevalence’? It's just a prevalence! In other samples, e.g., in other seasons or areas, it may be higher or lower. BUT here you are giving data for the particular sample studied. Why do you think that this level is "minimal"? Following your logic, you should then write ‘minimal mean intensity’. Correct it to ‘prevalence’.
Response: Thank you, this has been changed.
- Line 162.The scientific name of the host species should be given in Latin in the taxonomic summary here and below.
Response: Thank you, this has been corrected.
- Line 169. Remove "minimal" here and below, leaving only "prevalence".
Response: Thank you, this has been removed.
- Lines 190-192. Confusion with reference numbers. If I read correctly, it is a reference to an unpublished article and a comparison with an unpublished species. The only thing left is to believe the authors.
Response: The reference numbers have been corrected. The article was accepted for publication; its DOI is now also provided. The article being in press, hopefully it is going to be published soon.
- Species descriptions are very brief. Usually, the structures of the haptor are described first and then the sexual system.
Response: We have applied this, as the use of haptoral and genital hard parts in describing African dactylogyrids (and hence also the application of a fixative that allows to study these, and the production of drawings that show these) allow species classification in a higher-order taxon, or to diagnose specimens to species level. This has been set accordingly.
- The "Taxonomic discussion" does not correspond to "discussion", it is very brief and more in the spirit of "Differential diagnosis".
Only qualitative, descriptive characters are used for species differentiation.
It is desirable to extend the differentiation by comparing the metric characters that differ significantly.
Response: Qualitative characters are more specific than metric ones As most of metric characters overlap. We have then changed “Taxonomic discussion” into “Differential diagnosis”.
- Throughout the text, make sure that the Latin names of the species are in italics.
Response: Thank you, this has been verified.
- Lines 354-360 repeat lines 362-369. Remove the repetition.
Response: Thank you, the repetition has been removed.
Round 2
Reviewer 1 Report
Comments and Suggestions for Authors
- In the material and methods I didn't see the n of the samples. The work is classic and simple, but valid as long as it is well described.
Response: The sampling was opportunistic, so we did not have a target number of fishes to be dissected, in advance. Hence, the number of fishes sampled (dissected) has been mentioned in the Table 1 in the Results section (for each fish species, per locality)..
This study must be submitted to an ethics committee.
Author Response
Comments 1: In the material and methods I didn't see the n of the samples. The work is classic and simple, but valid as long as it is well described.
Response: The sampling was opportunistic, so we did not have a target number of fishes to be dissected, in advance. Hence, the number of fishes sampled (dissected) has been mentioned in the Table 1 in the Results section (for each fish species, per locality)..
This study must be submitted to an ethics committee.
Thank you, we provided to the Editor some documents, which are both authorizations for sampling, and the ethical approval for sampling, allowing to work on organisms (live specimen).
Round 3
Reviewer 1 Report
Comments and Suggestions for Authors
It was investigated the monopisthocotylan parasite fauna belonging to Quadriacanthus of five clariid fishes in the Upper Congo Basin. This sample should be much larger, at least three times, so the work should be continued.
Author Response
the number of hosts examined (between 2 and 12 per host species per location, with always several locations per host species) is not low at all for such a study. The number of parasites studied is obviously out of our control, as it depends on the number of parasites retrieved from the studied hosts. One of the three studied species is indeed, as we indicated in the manuscript, a rare species occurring at very low intensities. There is not much anyone can do about this - at such low infection rates, it would be unethical to continue sacrificing animals in sampling in the hope of obtaining more specimens. The number of parasites studied for the other two parasite species (37, and 49) is actually quite high for systematic parasitological research.